# Monetary incentives and peer referral in promoting secondary distribution of HIV self-testing among men who have sex with men in China: A randomized controlled trial

Yi Zhou[1,2]☯, Ying Lu[3,4]☯, Yuxin Ni[3,4]☯, Dan Wu[4,5], Xi He[6], Jason J. Ong[4,7,8], Joseph D. Tucker[4,5,9], Sean Y. Sylvia[10], Fengshi Jing[4,11], Xiaofeng Li[1], Shanzi Huang[1], Guangquan Shen[3,4], Chen Xu[3,4], Yuan Xiong[3,4], Yongjie Sha[3,4], Mengyuan Cheng[3,4], Junjie Xu[12], Hongbo Jiang[13], Wencan Dai[1], Liqun Huang[1], Fei Zou[14], Cheng Wang[3], Bin Yang[3], Wenhua Mei[1‡], Weiming Tang[3,4‡]*

1 Zhuhai Center for Disease Control and Prevention, Zhuhai, China, 2 Faculty of Medicine, Macau University of Science and Technology, Macau SAR, China, 3 Dermatology Hospital of South Medical University, Guangzhou, China, 4 University of North Carolina at Chapel Hill Project-China, Guangzhou, China, 5 Faculty of Infectious and Tropical Diseases, London School of Hygiene and Tropical Medicine, London, United Kingdom, 6 Zhuhai Xutong Voluntary Services Center, Zhuhai, China, 7 Central Clinical School, Faculty of Medicine, Monash University, Melbourne, Australia, 8 Melbourne Sexual Health Centre, Alfred Health, Melbourne, Australia, 9 Institute for Global Health and Infectious Diseases, School of Medicine, University of North Carolina at Chapel Hill, North Carolina, United States of America, 10 Department of Health Policy and Management, Gillings School of Global Public Health, University of North Carolina at Chapel Hill, Chapel Hill, North Carolina, United States of America, 11 School of Data Science, City University of Hong Kong, Hong Kong SAR, China, 12 Peking University Shenzhen Hospital, Shenzhen, China, 13 Guangdong Pharmaceutical University, Guangzhou, China, 14 Department of Biostatistics, Gillings School of Global Public Health, University of North Carolina at Chapel Hill, Chapel Hill, North Carolina, United States of America

☯ These authors contributed equally to this work.
‡ WM and WT also contributed equally to this work.
* Weiming_tang@med.unc.edu

## Abstract

### Background

Digital network–based methods may enhance peer distribution of HIV self-testing (HIVST) kits, but interventions that can optimize this approach are needed. We aimed to assess whether monetary incentives and peer referral could improve a secondary distribution program for HIVST among men who have sex with men (MSM) in China.

### Methods and findings

Between October 21, 2019 and September 14, 2020, a 3-arm randomized controlled, single-blinded trial was conducted online among 309 individuals (defined as index participants) who were assigned male at birth, aged 18 years or older, ever had male-to-male sex, willing to order HIVST kits online, and consented to take surveys online. We randomly assigned index participants into one of the 3 arms: (1) standard secondary distribution (control) group (*n* = 102); (2) secondary distribution with monetary incentives (SD-M) group (*n* = 103); and (3) secondary distribution with monetary incentives plus peer referral (SD-M-PR) group (*n* =

**Data Availability Statement:** All relevant data are within the manuscript and its Supporting Information files.

**Funding:** Financial Disclosure: This work was supported by the National Nature Science Foundation of China [grant numbers 81903371, WT], the National Key Research and Development Program of China [grant number 2017YFE0103800, WT], Academy of Medical Sciences and the Newton Fund [grant number NIF \R1\181020, DW], the National Institutes of Health [R34MH119963 (WT), R25 AI140495 (JT)]; and Zhuhai Medical and Health Science and Technology Plan Project [grant number 20181117A010064, YZ]. The funders had no role in study design, data collection and analysis, decision to publish, or preparation of the manuscript.

**Competing interests:** The authors have declared that no competing interests exist.

**Abbreviations:** CBO, community-based organization; CDC, Center for Diseases Control and Prevention; CHEERS, Consolidated Health Economic Evaluation Reporting Standards; CONSORT, Consolidated Standards of Reporting Trial; COVID-19, Coronavirus Disease 2019; HIVST, HIV self-testing; IRR, incidence rate ratio; LMIC, low- and middle-income country; MD, mean difference; MSM, men who have sex with men; OR, odds ratio; SD, standard deviation; SD-M, secondary distribution with monetary incentives; SD-M-PR, secondary distribution with monetary incentives plus peer referral; UNAIDS, Joint United Nations Programme on HIV/AIDS.

104). Index participants in 3 groups were encouraged to order HIVST kits online and distribute to members within their social networks. Members who received kits directly from index participants or through peer referral links from index MSM were defined as alters. Index participants in the 2 intervention groups could receive a fixed incentive ($3 USD) online for the verified test result uploaded to the digital platform by each unique alter. Index participants in the SD-M-PR group could additionally have a personalized peer referral link for alters to order kits online. Both index participants and alters needed to pay a refundable deposit ($15 USD) for ordering a kit. All index participants were assigned an online 3-month follow-up survey after ordering kits. The primary outcomes were the mean number of alters motivated by index participants in each arm and the mean number of newly tested alters motivated by index participants in each arm. These were assessed using zero-inflated negative binomial regression to determine the group differences in the mean number of alters and the mean number of newly tested alters motivated by index participants. Analyses were performed on an intention-to-treat basis. We also conducted an economic evaluation using microcosting from a health provider perspective with a 3-month time horizon. The mean number of unique tested alters motivated by index participants was 0.57 ± 0.96 (mean ± standard deviation [SD]) in the control group, compared with 0.98 ± 1.38 in the SD-M group (mean difference [MD] = 0.41),and 1.78 ± 2.05 in the SD-M-PR group (MD = 1.21). The mean number of newly tested alters motivated by index participants was 0.16 ± 0.39 (mean ± SD) in the control group, compared with 0.41 ± 0.73 in the SD-M group (MD = 0.25) and 0.57 ± 0.91 in the SD-M-PR group (MD = 0.41), respectively. Results indicated that index participants in intervention arms were more likely to motivate unique tested alters (control versus SD-M: incidence rate ratio [IRR = 2.98, 95% CI = 1.82 to 4.89, *p*-value < 0.001; control versus SD-M-PR: IRR = 3.26, 95% CI = 2.29 to 4.63, *p*-value < 0.001) and newly tested alters (control versus SD-M: IRR = 4.22, 95% CI = 1.93 to 9.23, *p*-value < 0.001; control versus SD-M-PR: IRR = 3.49, 95% CI = 1.92 to 6.37, *p*-value < 0.001) to conduct HIVST. The proportion of newly tested testers among alters was 28% in the control group, 42% in the SD-M group, and 32% in the SD-M-PR group. A total of 18 testers (3 index participants and 15 alters) tested as HIV positive, and the HIV reactive rates for alters were similar between the 3 groups. The total costs were $19,485.97 for 794 testers, including 450 index participants and 344 alter testers. Overall, the average cost per tester was $24.54, and the average cost per alter tester was $56.65. Monetary incentives alone (SD-M group) were more cost-effective than monetary incentives with peer referral (SD-M-PR group) on average in terms of alters tested and newly tested alters, despite SD-M-PR having larger effects. Compared to the control group, the cost for one more alter tester in the SD-M group was $14.90 and $16.61 in the SD-M-PR group. For newly tested alters, the cost of one more alter in the SD-M group was $24.65 and $49.07 in the SD-M-PR group. No study-related adverse events were reported during the study. Limitations include the digital network approach might neglect individuals who lack internet access.

## Conclusions

Monetary incentives alone and the combined intervention of monetary incentives and peer referral can promote the secondary distribution of HIVST among MSM. Monetary incentives can also expand HIV testing by encouraging first-time testing through secondary distribution by MSM. This social network–based digital approach can be expanded to other public health research, especially in the era of the Coronavirus Disease 2019 (COVID-19).

## Trial registration

Chinese Clinical Trial Registry (ChiCTR) ChiCTR1900025433

Author summary

### Why was this study done?

- Men who have sex with men (MSM) in China have a high burden of HIV, while testing coverage remains low.

- HIV self-testing (HIVST) is an effective approach to supplement HIV testing services and engage marginalized individuals who have avoided facility-based testing due to potential stigma and discrimination.

- The strategy of secondary distribution, whereby individuals receive multiple HIVST kits and distribute them to members within their social networks, including sexual partners and friends, has been adopted by some HIVST programs and proved it can improve HIV testing coverage.

- We aimed to further innovate and promote the current secondary distribution model to a digital-based one among MSM in China, where they can order HIVST online. We also added monetary incentives and peer referral interventions to evaluate whether they can amplify the effectiveness of secondary distribution.

### What did the researchers do and find?

- We recruited 309 participants to evaluate the effect of 2 interventions: (1) monetary incentives group; and (2) monetary incentives plus peer referral group in promoting the secondary distribution of HIVST compared to the control group, between October 2019 and September 2020.

- In the 2 intervention groups, the mean number of tested alters motivated by index participants was 0.98 (SD = 1.38) in the monetary incentives group and 1.78 (SD = 2.05) in the monetary incentives plus peer referral group, compared to 0.57 (SD = 0.96) in the control group.

- The average cost per alter tested was $61.58 in the monetary incentives group and $41.56 in the monetary incentives plus peer referral group, compared to $96.18 in the control group.

### What do these findings mean?

- Both monetary incentives and monetary incentives plus peer referral can effectively engage more individuals to conduct HIVST and increase the HIV testing uptake among MSM.

- The interpretation of our study's findings might not apply to marginalized individuals who have limited access to the internet, and, therefore, future studies are needed to address this limitation.

## Introduction

The Coronavirus Disease 2019 (COVID-19) pandemic has seriously impacted the AIDS response [1]. Achieving the Joint United Nations Programme on HIV/AIDS (UNAIDS) 2025 targets will require openness to innovation and scale-up of breakthroughs in technology and service delivery to put people at the center, especially among marginalized populations [2]. These response strategies would also be relevant to other global public health issues. Men who have sex with men (MSM) in China have a high burden of HIV, while testing coverage remains low. In 2019, HIV prevalence was 6.3% among MSM, but only 56.4% were aware of their HIV status [3]. Thus, methods that can increase HIV testing yield are needed. COVID-19–related restrictions have closed many HIV clinical services, further increasing barriers to HIV testing [1,4].

HIV self-testing (HIVST) has been shown as an effective approach to supplement the HIV testing services and reach individuals who have avoided facility-based testing [5]. Both during the COVID-19 pandemic and in the long run, HIVST can address unique programmatic needs and global HIV testing targets [6]. Particularly, in China, studies suggested that HIV phobia and institutional discrimination against people living with HIV are common [7]. The confidential and private nature of HIVST have the potential for addressing stigma and discrimination. More importantly, HIVST as an innovative method can be expected to reach and link people to care [8]. Social network–based HIV testing approaches are an extension of HIV partner services, where individuals encourage their sexual or social contacts to conduct HIV tests [9]. They are effective methods in optimizing HIV care and prevention [10]. South African and Ugandan studies showed that peer-driven distribution of HIVST within sexual networks can expand HIV test uptake among MSM [11,12]. Partner distribution of HIVST has increased partner testing among HIV–negative female sex workers and women seeking antenatal and postpartum care in Kenya [13,14]. HIVST secondary distribution is a social network–based method whereby individuals receive multiple HIVST kits and distribute them to people within their social networks, including sexual partners and friends [13]. Secondary distribution can improve HIV testing coverage. However, the effectiveness of current social network methods is still limited, stressing the necessity to innovate the strategies [13–15].

Monetary incentives can increase demand for self-tests and work synergistically alongside network methods to increase HIV testing uptake[16,17]. Conditional economic incentives refer to the financial or other material rewards given to participants for performing specific behaviors that may effectively increase HIV testing in low- and middle-income countries (LMICs) [17]. In the case of self-testing and distribution to sexual partners, insights from psychology and behavioral economics suggest that the "present-biased preferences" of individuals or preference for present smaller rewards overtakes the larger benefits in the future, maybe a barrier to uptake [18]. This occurs in HIVST where there are immediate logistical and psychological costs, but the benefits of self-testing are realized later [19]. The effectiveness of secondary distribution programs, on the other hand, may depend more on intrinsic prosocial preferences (or altruistic motives) of those distributing kits to peers [20], and monetary incentives could backfire. Thus, extrinsic monetary incentives could either enhance or diminish intrinsic motivation to distribute kits [21,22].

In-person delivery is one of the limitations of most HIVST secondary distribution models that increase barriers for the index participants to distribute. Online peer referrals can be a solution to this problem. MSM can refer members of their social networks online to order HIVST kits without any offline interactions. Peer referral leverages social networks in identifying undiagnosed HIV infections among marginalized and key populations [12,19,23–25]. By taking advantage of both social networks and digital platforms, peer referral with monetary incentives may enhance the effectiveness of HIV testing through a digital network–based

HIVST service. In addition, peer-based referral systems may leverage some of the trust built into the social network of index MSM and motivate people to test [26,27].

This study aimed to evaluate the effectiveness and cost-effectiveness of monetary incentives and peer referral in promoting the digital network–based secondary distribution of HIVST among MSM in China using a 3-arm randomized controlled trial.

## Methods

### Study design and participants

The study was conducted jointly by the University of North Carolina at Chapel Hill Project-China, Zhuhai Center for Diseases Control and Prevention (CDC), and Zhuhai Xutong Voluntary Services Center (Xutong). Xutong is a gay-led community-based organization (CBO) in Zhuhai. The study utilized an HIVST online ordering system developed by Xutong. The ordering system was hosted and managed using WeChat, China's largest social networking platform.

In this study, we had 3 study groups: (1) standard secondary distribution (control) group; (2) secondary distribution with monetary incentives (SD-M) group; and (3) secondary distribution with monetary incentives plus peer referral (SD-M-PR) group. The trial procedures were similar for the 3 groups, all of which included a secondary distribution process. What distinguished the control group and intervention groups were that monetary incentives were implemented in the SD-M group and in the SD-M-PR group. In addition to monetary incentives, online peer referral, which was a virtual secondary distribution method, was added in the SD-M-PR group. We planned to recruit 300 eligible participants (100 for each arm) with the following inclusion criteria: (1) assigned male at birth; (2) aged 18 years or older; (3) ever had male-to-male sex; (4) be willing to order HIVST kits online; (5) be consented to take baseline and follow-up surveys online; and (6) ordered HIVST through the online platform. We assessed testing uptake rates and administered a follow-up survey 3 months after index participants ordering. All participants gave digital written informed consent by providing electronic signatures before the online baseline survey. Ethics approval was obtained through the Zhuhai CDC, and the study protocol of the trial was published [28].

### Randomization and masking

Volunteers of Xutong enrolled all the participants by advertising the recruitment and trial introduction in the WeChat public platform. Eligible participants were randomly assigned to one of the 3 arms individually and independently by a computer-generated program electronically without the need for staff interaction. Index participants were asked to play a spinning game at the beginning of the baseline survey, which was to assign them into one of the groups randomly based solely on a 1:1:1 allocation ratio. Each participant can only play the game once based on the IP address and phone number. Both study staff and index participants knew about the group assignment at the same time when index participants submitted the baseline survey. Both study staff and index participants were unblinded to allocation, and they knew about the group assignment at the same time when index participants completed the baseline survey. The group allocations remained concealed as the randomization took place before the interventions, and participants could not switch study arms later. Only the statisticians who analyzed the data remained blind to participant allocations.

### Procedures

**Control group: Standard of care.** The control group was a standard secondary distribution care model developed and implemented by Xutong [15]. Index participants in the control

group completed the baseline survey and ordered HIV/Syphilis dual rapid test kits (SD Bioline, South Korea) through the online system. Index participants ordered HIVST kits by paying a refundable deposit for each kit, 100 RMB ($15), which was the equal value of the testing kit and gifts (condoms and lubricants) included in the testing package. Each HIVST kit was also attached with a self-testing instruction and a "returning card." A returning card contained a QR code that allowed testers to scan and upload the photos of the test results anonymously and privately. There was also a confirmation number in the returning card to verify different index participants and their alters. Each index participant shared the same confirmation code with their alters. HIVST kits, self-test instructions, gifts, and returning cards were packaged in unlabeled boxes and mailed to index participants after they provided the shipping addresses and paid the deposit. Index participants could order up to 5 HIVST kits and arrange them by themselves by conducting self-tests, testing together with alters, or distributing the kits to alters.

After testing, all testers scanned the QR code in the returning card to submit photographs of their test results and reported whether they were index participants or alters. Alter testers also completed an anonymous survey. Index and alter testers were encouraged to upload their testing results separately and privately using the returning card and did not need to share the results. Testing was free for both index participants and alters, and trained CBO staff refunded all deposits via online transaction after verifying submitted test results. Those who tested positive (reactive) for HIV were referred for confirmatory testing at the local CDC and provided other counseling and linkage to care services by peer volunteers.

**SD-M group: Secondary distribution with monetary incentives.**   Index participants in this arm received the same procedures as the control group, with monetary incentives. Index participants could receive monetary incentives for self-tests and distribution. They could receive 20 RMB (approximately equal to $3) for self-tests. In terms of distribution, both the index participant and the alter could receive a 20 RMB when self-test results were uploaded and verified by each unique alters (by checking the confirmation code and phone number). All incentives were through online transactions. In this group, index participants could refer up to 5 alters and receive a maximum of 100 RMB incentives.

**SD-M-PR group: Secondary distribution with monetary incentives plus peer referral.** Participants in this arm were provided with the option of online peer referral, in addition to the monetary incentive arm procedures as previously described. Except for ordering and distributing kits directly to the alters, index participants in this arm could also utilize the approach of peer referral to complete the secondary distribution virtually. Index participants received a unique and personalized peer referral link to share with the alters, and they could distribute up to 5 HIVST kits through this channel (each alter could order 1 test kit through the link). In the system of peer referral, referred alters clicked the link shared by the index participant and completed the process of ordering and returning all by themselves. Alters paid the refundable deposit of 100 RMB ($15) online, provided the address, received the testing kits, conducted self-tests, and uploaded the results back to the platform. Similarly, both index participants and alters received a 20 RMB ($3) for verified tests by a unique alter. In this group, index participants could refer to up to 10 alters (5 kits distribution + 5 peer referral) and receive a maximum of 200 RMB.

## Outcomes

We define a motivated alter as an individual who conducted HIVST using the distributed kits from index MSM, uploaded the testing result, and the result was verified by trained CBO volunteers. For testers who uploaded results more than once, we removed duplication based on their phone numbers and used their most recent results for data analysis. Primary outcomes

were (1) the mean number of unique tested alters motivated by index participants; and (2) the mean number of newly tested alters motivated by index participants for each arm. Secondary outcomes included the mean number of alters who tested positive for HIV recruited by index participants in each arm; the mean number of alters who tested positive for syphilis recruited by index participants in each arm; and evaluation of intervention effects for the mean number of alters motivated by index participants in subgroups defined by age, residence, sexual orientation, sexual behavior, and previous testing experience. Using micro-costing, we estimated the costs, including fixed capital costs, fixed costs for staff, fixed consumable costs, variable staff costs, and variable consumable costs, per person tested and costs per alter and newly tested alter of SD-M and SD-M-PR versus the control group. Specific amendments and rationale to primary outcomes and methods of outcome analyses were described in supporting materials (S4 File).

## Statistical analysis

We calculated the study sample size based on our pilot study findings that the mean number of alters motivated by index participants was 0.65 by standard secondary distribution, 1.0 by SD-M, and 1.4 by secondary distribution with monetary incentives plus peer referral, respectively [15]. Assuming equal variance among the 3 groups, we applied a standard deviation (SD) of 0.5 (preliminary results) for all groups. A minimum detectable size of 0.35 would be of interest. Assuming a 20% loss to follow-up, 300 participants, 100 in each group, would be required to provide 90% power with an alpha of 0.05.

Baseline characteristics of index participants were reported using descriptive statistics. We used chi-squared tests to compare baseline differences between index participants who completed the 3-month follow-up survey and those who did not. A $p$-value is less than 0.05 was deemed statistically significant. We used zero-inflated negative binomial regression for primary outcomes analyses. More specifically, the first primary outcome analyses compared the mean number of unique alters motivated by index participants between the intervention groups (SD-M and SD-M-PR) and the control group and between the 2 intervention groups (SD-M and SD-M-PR) by estimating the incidence rate ratio (IRR) and its associated 95% CI. The second primary outcome analysis compared the mean number of newly tested alters motivated by index participants by estimating IRR and its associated 95% CI. We applied logistic regression for secondary outcome analyses, i.e., the mean number of alters with HIV reactive results identified by index participants, by estimating the odds ratio (OR) and its associated 95% CI. We conducted a subgroup analysis for the first primary outcome stratified by age (over or under 30 years), sexual orientation (self-identified as gay or others), previous testing experience (ever tested for HIV or not), or sexual behavior (had condomless anal sex in the past 3 months or not). All IRRs and ORs were adjusted by index participants' age, highest education level, income, residence, marital status, and sexual orientation. All statistical analyses were performed using R Studio, Version 1.2.5033.

We conducted an economic evaluation alongside the trial by using microcosting to estimate the total cost of each trial arm, the cost per person tested, the cost per alter tested, and the cost per alter diagnosed with HIV. Fixed (e.g., building rental and management expenses, office equipment, and capacity building) and variable costs (e.g., consumables, personnel time, and incentive cost) were collected from a healthcare provider perspective with a time horizon of 3 months. Economic costs were converted from Chinese Yuan (RMB) and presented as US Dollars USD (2020) (1 RMB = $0.15 USD).

We followed the intention-to-treat principle and did not have missing data in terms of primary and secondary outcomes. We did not have a data monitoring committee because the

potential trial risks were minor. This study was registered in the Chinese Clinical Trial Registry (ChiCTR1900025433). Results reporting followed the extension of the Consolidated Standards of Reporting Trial (CONSORT) 2010 statement [29] (S1 File). The economic analysis was reported according to the Consolidated Health Economic Evaluation Reporting Standards (CHEERS) statement [30] (S2 File).

## Results

We assessed 417 MSM for eligibility and recruited 309 for inclusion in the study from October 21, 2019 to September 14, 2020. One participant (<1%, 1/417) did not sign the informed consents, 89 (21%, 89/417) consented but did not pay the deposit for unknown reasons, and 18 (4%, 18/417) were excluded with reasons due to unexpected technical issues or study inclusion criteria. A total of 309 MSM (74%, 309/417) from 68 cities in China were recruited and randomized into control ($n$ = 102), SD-M ($n$ = 103), and SD-M-PR ($n$ = 104) groups, respectively (Fig 1).

After 3 months, 96 people in the control group, 97 people in the SD-M group, and 100 people in the SD-M-PR arm completed the follow-up survey, with follow-up rates of 94% (96/102), 94% (97/103), and 96% (100/104), respectively. There was no significant difference ($p$-value > 0.05) in the social demographic characteristics between index participants who completed the 3-month follow-up survey and those who did not (S1 Table).

Table 1 provides the baseline characteristics of enrolled index participants, including their social demographics, sexual behaviors, and HIV testing history. All baseline variables did not follow a normal distribution. Index participants shared similar baseline characteristics among the 3 groups. The mean age was 30 years. A total of 70% had attended college, 72% self-identified as gay, 84% identified as single, 35% reported condomless anal sex within the past 3 months, and 85% had tested for HIV before.

Fig 1 shows the total number of kits ordered and secondary distributed by index participants within each arm. Index participants in the control arm ordered a total of 222 kits, and testers returned 209 test results (returning rate: 94%, 209/222), among which 144 results were from index participants self-testing, and 65 (31%, 65/209) results were from 58 unique alters. In the SD-M arm, a total of 275 kits were ordered by the index participants, and we received 257 (returning rate: 93%, 257/275) test results, among which 150 results were from index participants self-testing, and 107 (42%, 107/257) results were from 101 unique alters. In the SD-M-PR arm, index participants ordered 262 kits and referred 75 links, and we received a total of 328 (returning rate: 97%, 328/337) test results, among which 141 results were from index participants self-testing, and 187 (57%, 187/328) results were from 185 unique alters including 110 alters through offline kits distribution, and 75 alters through the online peer referral link. The discrepancies between the number of used kits and the number of identified alters (control: 65 versus 58; SD-M: 107 versus 101; SD-M-PR: 187 versus 185) are because of alters who tested and uploaded results more than once. More information regarding the number of kits ordered and actual kits distribution by index in each group is listed in S2 Table. The number of index participants who successfully motivated alters to HIVST was 40 (39%, 40/102) in the control group, 49 (48%, 49/103) in the SD-M, and 51 (49%, 51/104) in the SD-M-PR, respectively. In addition, each index participant who successfully motivated alters to HIVST ordered 2.9 kits on average in the control group, 3.6 kits in the SD-M group, and 3.5 kits in the SD-M-PR group (excluding peer referral links). Total 8 alters (7 in the control arm and 1 in the SD-M arm) did not want to complete the survey for alters and self-reported as index participants when returning results, according to the feedback from CBO staff. Most alters were same-sex partners (37%, 97/265) or gay friends (52%, 139/265) of the index

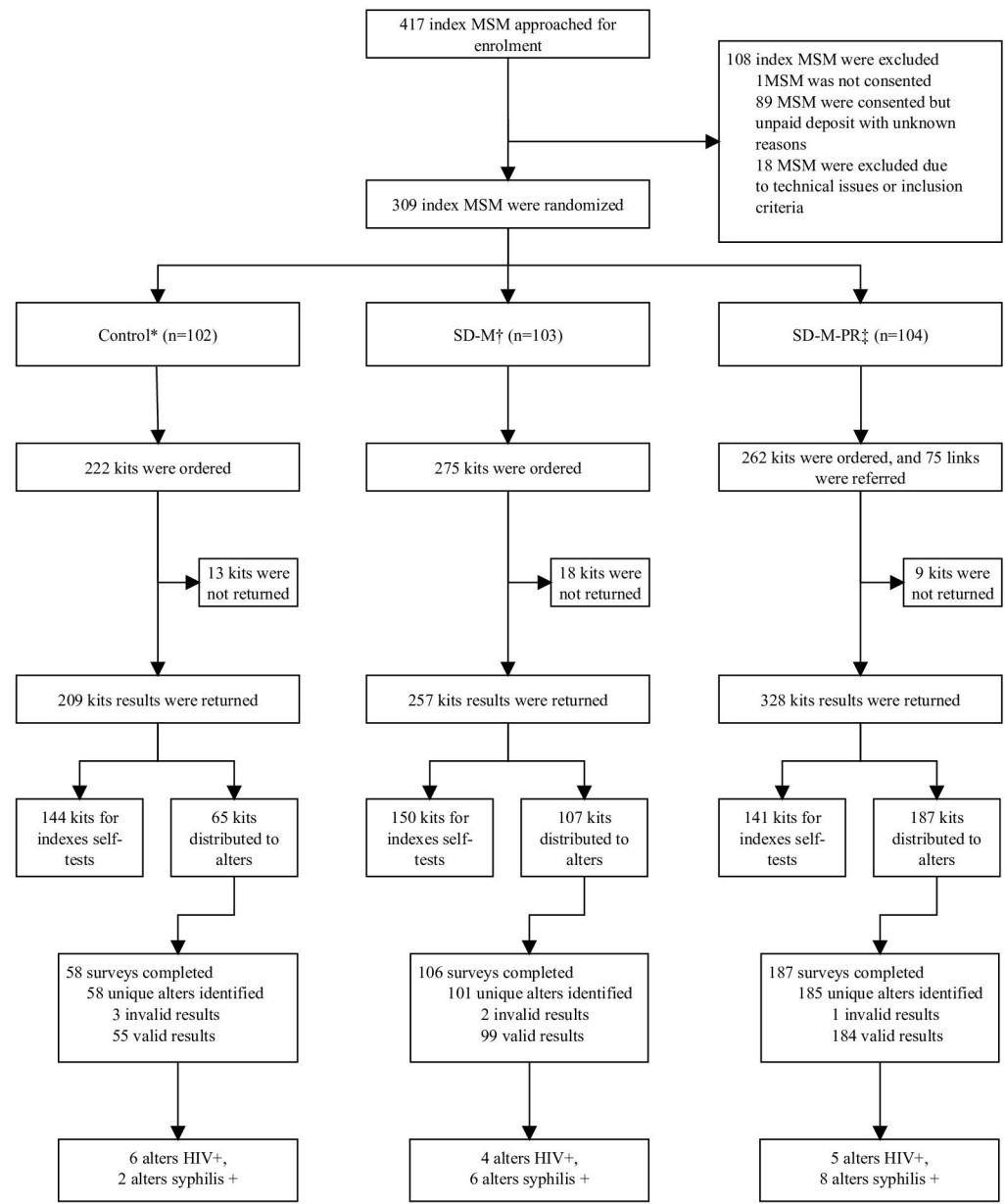

The number of index participants analysed in the intention-to-treat was 102 in the control group, 103 in the SD-M group, and 104 in the SD-M-PR group.

*Control refers to standard secondary distribution group.
†SD-M refers to secondary distribution with monetary incentives group.
‡SD-M-PR refers to secondary distribution with monetary incentives plus peer referral group.

**Fig 1. Trial profile.** MSM, men who have sex with men; SD-M, secondary distribution with monetary incentives; SD-M-PR, secondary distribution with monetary incentives plus peer referral.

participants, according to 269 alters who completed returned survey. More information regarding alters' characteristics is listed in S3 Table.

Table 2 shows the primary outcome analyses with a total of 309 index participants and 334 unique alters. In the control group, the mean number of tested alters motivated by index

**Table 1. Baseline characteristics of the index participants in Zhuhai, China, 2019 to 2020 (N = 309).**

| | Control* n = 102 | SD-M† n = 103 | SD-M-PR‡ n = 104 | Overall n = 309 |
|---|---|---|---|---|
| Age (years) | 29.76 (6.02) | 29.83 (7.21) | 29.75 (6.48) | 29.75 (6.57) |
| ≤ 30 | 66 (65%) | 67 (65%) | 69 (66%) | 202 (65%) |
| > 30 | 36 (35%) | 36 (35%) | 35 (34%) | 107 (35%) |
| Education | | | | |
| High school or below | 20 (20%) | 21 (20%) | 21 (20%) | 62 (20%) |
| College | 70 (69%) | 76 (74%) | 75 (72%) | 221 (72%) |
| Master's degree or above | 12 (12%) | 6 (6%) | 8 (8%) | 26 (8%) |
| Monthly Income (USD, $) | | | | |
| <225 | 8 (8%) | 8 (8%) | 8 (8%) | 24 (8%) |
| 225–450 | 7 (7%) | 7 (7%) | 5 (5%) | 19 (6%) |
| 451–750 | 16 (16%) | 34 (33%) | 24 (23%) | 74 (24%) |
| 751–1200 | 37 (36%) | 27 (26%) | 39 (38%) | 103 (33%) |
| >1200 | 34 (33%) | 27 (26%) | 28 (27%) | 89 (29%) |
| Sexual Orientation | | | | |
| Gay | 73 (72%) | 68 (66%) | 80 (77%) | 221 (72%) |
| Others§ | 29 (28%) | 35 (34%) | 24 (23%) | 88 (28%) |
| Sexual Orientation Disclosure‖ | | | | |
| Yes | 57 (56%) | 58 (56%) | 55 (53%) | 170 (55%) |
| No | 45 (44%) | 45 (44%) | 49 (47%) | 139 (45%) |
| Marital Status | | | | |
| Single | 87 (85%) | 87 (85%) | 87 (84%) | 261 (84%) |
| Engaged or married | 14 (14%) | 10 (10%) | 13 (13%) | 37 (12%) |
| Separated or divorced | 1 (1%) | 6 (6%) | 4 (4%) | 11 (4%) |
| Residence | | | | |
| Guangdong province | 73 (72%) | 71 (69%) | 67 (64%) | 211 (68%) |
| Other provinces | 29 (28%) | 32 (31%) | 37 (36%) | 98 (32%) |
| Condomless sex in the past 3 months# | | | | |
| Yes | 35 (34%) | 39 (38%) | 33 (32%) | 107 (35%) |
| No | 67 (66%) | 64 (62%) | 71 (68%) | 202 (65%) |
| Ever tested for HIV | | | | |
| Yes | 87 (85%) | 89 (86%) | 86 (83%) | 262 (85%) |
| No | 15 (15%) | 14 (14%) | 18 (17%) | 47 (15%) |

Age data are presented as mean (SD). All other data are presented as *n* (%).

*Control refers to standard secondary distribution group.

†SD-M refers to secondary distribution with monetary incentives group.

‡SD-M-PR refers to secondary distribution with monetary incentives plus peer referral groups.

§Others include heterosexual, bisexual, and not sure.

‖Sexual orientation disclosure refers to whether an index discloses his sexual orientation to people other than sexual partners, e.g., healthcare providers, friends, family members, etc.

#Condomless sex in the past 3 months refers to whether an index had condomless sex with another man in the past 3 months.

SD, standard deviation.

participants was 0.57 (SD = 0.96), compared with 0.98 (SD = 1.38) in SD-M with a mean difference (MD) of 0.41 and 1.78 (SD = 2.05) in SD-M-PR with an MD of 1.21, respectively. Compared with index participants in the control group, index participants in intervention groups were more likely to motivate more unique alters to self-tested for HIV (control versus SD-M: IRR = 2.98, 95% CI = 1.82 to 4.89, *p*-value < 0.001; control versus SD-M-PR: IRR = 3.26, 95%

**Table 2. Outcome analysis of 309 index participants in China at the end of the study, 2019 to 2020.**

| Outcomes | | Unique alters motivated by index participants | Newly-tested alters motivated by index participants | Alters with HIV-reactive results identified by index participants |
|---|---|---|---|---|
| **Groups, n (Mean, SD)** | Control* (N = 102) | 58 (0.57, 0.96) | 16 (0.16, 0.39) | 6 (0.06, 0.24) |
| | SD-M† (N = 103) | 101 (0.98, 1.38) | 42 (0.41, 0.73) | 4 (0.04, 0.19) |
| | SD-M-PR‡ (N = 104) | 185 (1.78, 2.05) | 59 (0.57, 0.91) | 5 (0.05, 0.17) |
| **Mean Difference** | Control (ref) vs SD-M | 0.41 | 0.25 | -0.02 |
| | Control (ref) vs SD-M-PR | 1.21 | 0.41 | -0.01 |
| | SD-M (ref) vs SD-M-PR | 0.80 | 0.16 | 0.01 |
| **Models, IRR/OR (95% CI), p-value** | Control (ref) vs SD-M | 2.98 (1.82, 4.89) *** | 4.22 (1.93, 9.23) *** | 0.79 (0.18, 3.27) |
| | Control (ref) vs SD-M-PR | 3.26 (2.29, 4.63) *** | 3.49 (1.92, 6.37) *** | 0.79 (0.21, 2.89) |
| | SD-M (ref) vs SD-M-PR | 1.25 (0.87, 1.81) | 1.45 (0.58, 3.61) | 1.19 (0.25, 6.24) |

Mean refers to each outcome/mean number of index participants in each group.

*Control refers to the standard secondary distribution group.

†SD-M refers to secondary distribution with monetary incentives group.

‡SD-M-PR refers to secondary distribution with monetary incentives plus peer referral group.

ZINB was used to calculate IRR for outcomes "the total number of unique alters motivated by index participants" and "the total number of newly tested alters motivated by index participants." Logistic regression was used to calculate OR for outcome "the total number of alters with HIV reactive results identified by index participants."

IRR or OR were all adjusted for age, income, education, marital status, sexual orientation, and residence.

***$p < 0.001$.

IRR, incidence rate ratio; MD, mean difference; OR, odds ratio; SD, standard deviation; ZINB, zero-inflated negative binomial regression.

CI = 2.29 to 4.63, $p$-value < 0.001, respectively). The total number of unique alters motivated by index participants was not associated with the assignment of index participants to the SD-M group or the SD-M-PR group (IRR = 1.25, 95% CI = 0.87 to 1.81, $p$-value > 0.05).

Comparatively, the mean number of newly tested motivated by index participants was 0.16 (SD = 0.39) in the control group, compared with 0.41 (SD = 0.73) in SD-M with an MD of 0.25, and 0.57 (SD = 0.91) in SD-M-PR with an MD of 0.41. The likelihood that the total number of newly tested alters motivated by index participants was significantly increased when index participants were in the SD-M group or SD-M-PR group, compared with the one in the control group (control versus SD-M: IRR = 4.22, 95% CI = 1.93 to 9.23, $p$-value < 0.001; control versus SD-M-PR: IRR = 3.49, 95% CI = 1.92 to 6.37, $p$-value < 0.001, respectively). The total number of newly tested alters motivated by index participants was not associated with whether index participants were grouped in SD-M or SD-M-PR (IRR = 1.45, 95% CI = 0.58 to 3.61, $p$-value > 0.05).

Of the unique alters in each group (control group: 58; SD-M: 101; SD-M-PR: 185), 28% (16/58) of alters in the control group were newly tested, compared to 42% (42/101) of alters in the SD-M, and 32% (59/185) of alters in the SD-M-PR (Fig 2).

In total, 18 testers were diagnosed with HIV, including 3 index participants (2 newly diagnosed with HIV) and 15 alters (13 newly diagnosed). Of the 15 alters who were diagnosed with HIV, 6 were from the control group (5 newly diagnosed with HIV), 4 from SD-M (4 newly diagnosed with HIV), and 5 from SD-M-PR (4 newly diagnosed with HIV) (Fig 1). The total number of unique alters with HIV reactive results motivated by index participants was not associated with the assignment of index participants to the groups (Table 2).

We also identified 31 participants who tested positive for syphilis, including 15 index participants and 16 alters (2 from the control group, 6 from the SD-M group, and 8 from the SD-M-PR group).

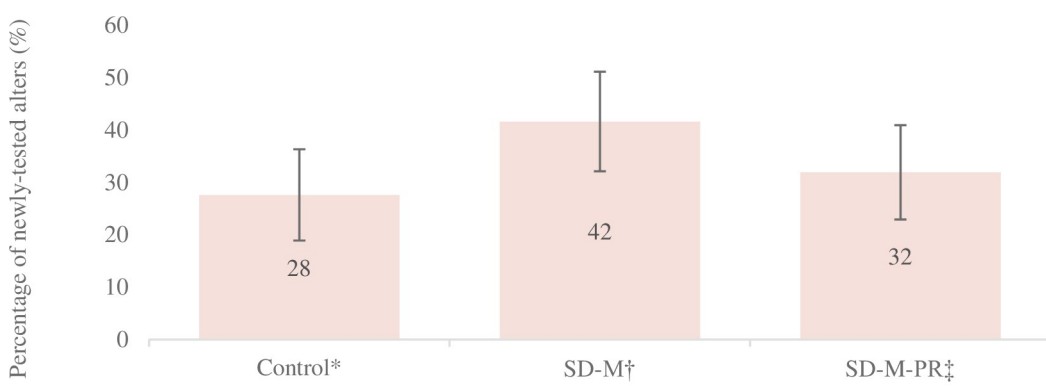

| | Control* | SD-M† | SD-M-PR‡ |
|---|---|---|---|
| Number of alters | 58 | 101 | 185 |
| Number of newly tested alters | 16 | 42 | 59 |

*Control refers to standard secondary distribution group. †SD-M refers to secondary distribution with monetary incentives group. ‡SD-M-PR refers to secondary distribution with monetary incentives plus peer referral group. Error bars are presented as 95% confidence interval.

**Fig 2. Percentage of newly tested alters among 344 alters at the end of the study, 2019 to 2020.** SD-M, secondary distribution with monetary incentives; SD-M-PR, secondary distribution with monetary incentives plus peer referral.

The effects of intervention groups varied according to index participants' age, self-identified sexual orientation, sexual behavior, and previous HIV testing experience. A higher number of motivated unique alters was associated with index participants in the intervention groups, aged over 30, who self-identified as gay, previously tested for HIV, and had condomless sex within 3 months. Compared to those in the control group, index participants (SD-M-PR group) who disclosed sexual orientation had a significantly positive association with the number of unique alters, while index participants in the SD-M group had a significant negative association (Table 3).

Table 4 summarizes the results of the economic evaluation. The total program costs were estimated to be $19,485.97 for 794 testers, including 450 index participants and 344 alter testers. Overall, the average cost per tester was $24.54, and the average cost per alter tester was $56.65. S4 Table and S1 and S2 Figs give further details of the fixed and variable costs for all 3 groups. The average cost per tester in the SD-M group was $24.20 compared with $23.44 in the SD-M-PR group and $26.69 per tester in the control group. For alter testers, the mean costs in the SD-M and SD-M-PR groups were $61.58 and $41.56, respectively, whereas that in the control group was $96.18. In addition, the average costs increased substantially to more than $100 for one newly tested alter in all 3 groups (Table 4). Compared to the control group, cost for one more alter tester in the SD-M group was $14.90 and $16.61 in the SD-M-PR group. The incremental cost per additional alter tested in the SD-M-PR group compared to the SD-M group was $17.49. For newly tested alters, the cost of one more alter in the SD-M group was $24.65 and was $49.07 in the SD-M-PR group, compared to the controls. Moving from SD-M to SD-M-R, the incremental cost per newly tested alter was $86.42. Both SD-M group and SD-M-PR group were dominated by the control group as they incurred higher costs yet produced fewer HIV reactive results.

There were no study-related adverse events reported within the trial duration.

**Table 3. Subgroup analysis of 309 index participants regarding the total number of motivated unique alters in China at the end of the study, 2019 to 2020.**

| Variables | | Models, IRR (95% CI) | | |
|---|---|---|---|---|
| | | Control[α] vs SD-M[†] | Control vs SD-M-PR[‡] | SD-M vs SD-M-PR |
| Age (years) | <30 | 1.82 (0.90, 3.69) | 2.92 (1.42, 6.01) ** | 1.47 (0.88, 2.46) |
| | ≥30 | 4.20 (2.09, 8.47) *** | 5.30 (2.59, 10.83) *** | 1.10 (0.60, 2.04) |
| Sexual orientation | Gay | 3.19 (1.81, 5.63) *** | 2.92 (1.37, 6.21) ** | 1.02 (0.65, 1.59) |
| | Others§ | 1.70 (0.85, 3.39) | 5.37 (2.78, 10.35) *** | 2.39 (1.33, 4.27) ** |
| Sexual orientation disclosure | Yes | 4.01 (2.35, 6.83) *** | 3.06 (1.98, 4.72) *** | 0.97 (0.65, 1.44) |
| | No | 1.00 (0.47, 2.12) | 4.09 (2.27, 7.38) *** | NA |
| Ever tested for HIV | Yes | 3.01 (1.78, 5.09) *** | 3.50 (1.87, 6.53) *** | 1.17 (0.80, 1.70) |
| | No | 0.78 (0.18, 3.30) | NA | 4.37 (1.42, 13.5) * |
| Condomless sex in the past 3 months# | Yes | 7.94 (3.53, 17.9) *** | 6.29 (3.16, 12.51) *** | 1.37 (0.84, 2.23) |
| | No | 1.41 (0.77, 2.59) | 2.53 (1.63, 3.91) *** | 1.52 (0.84, 2.76) |

[α]Control refers to standard secondary distribution group.

[†]SD-M refers to secondary distribution with monetary incentives group.

[‡]SD-M-PR refers to secondary distribution with monetary incentives plus peer referral group.

§Others include heterosexual, bisexual, and not sure.

#Condomless sex in the past 3 months refers to whether an index participant had condomless sex with another man in the past 3 months.

ZINB was used to calculate IRRs. NA was shown because of subsets of data did not meet assumptions of ZINB. IRRs were adjusted for age, income, education, marital status, sexual orientation, and residence.

*$p < 0.05$,

**$p < 0.01$,

***$p < 0.001$.

IRR, incidence rate ratio; ZINB, zero-inflated negative binomial regression.

**Table 4. Economic evaluation of the randomized controlled trial in China, 2019 to 2020.**

| | Control* | SD-M[†] | SD-M-PR[‡] | SD-M–Control (ref) | | SD-M-PR–Control (ref) | | SD-M-PR—SD-M (ref) | |
|---|---|---|---|---|---|---|---|---|---|
| | | | | Difference between groups | Incremental cost per outcome | Difference between groups | Incremental cost per outcome | Difference between groups | Incremental cost per outcome |
| **Total testers (index participants and alters)** | | | | | | | | | |
| Costs | 5578.44 | 6219.23 | 7688.30 | 640.79 | 13.35 | 2109.86 | 17.73 | 1469.08 | 20.69 |
| Outcomes | 209 | 257 | 328 | 48 | | 119 | | 71 | |
| **Unique alter testers** | | | | | | | | | |
| Costs | 5578.44 | 6219.23 | 7688.30 | 640.79 | 14.90 | 2109.86 | 16.61 | 1469.08 | 17.49 |
| Outcomes | 58 | 101 | 185 | 43 | | 127 | | 84 | |
| **Newly-tested alters** | | | | | | | | | |
| Costs | 5578.44 | 6219.23 | 7688.30 | 640.79 | 24.65 | 2109.86 | 49.07 | 1469.08 | 86.42 |
| Outcomes | 16 | 42 | 59 | 26 | | 43 | | 17 | |
| **Alters with HIV-reactive results** | | | | | | | | | |
| Costs | 5578.44 | 6219.23 | 7688.30 | 640.79 | Dominated | 2109.86 | Dominated | 1469.08 | Dominated |
| Outcomes | 6 | 4 | 5 | -2 | | -1 | | 1 | |

*Control refers to standard secondary distribution group.

[†]SD-M refers to secondary distribution with monetary incentives group.

[‡]SD-M-PR refers to secondary distribution with monetary incentives plus peer referral group.

## Discussion

Improving HIV testing uptake is an important strategy in response to the global AIDS challenges. Social network–based secondary distribution of HIVST is feasible and effective in promoting HIV test uptake and identifying new testers among social contacts of MSM. Our study aimed to further optimize the effectiveness of secondary distribution as an innovative HIVST strategy. Our findings indicated that both incentive and peer referral could improve the efficacy of secondary distribution of HIVST among Chinese MSM. This 3-arm randomized controlled trial extends the literature by comparing the cost evaluation of 2 enhanced secondary distribution strategies, leveraging social networks, and drawing on digital platforms.

Our trial extends knowledge on HIV care prevention in several ways by incorporating the use of digital distribution and incentives into an existing MSM community–led, social network–based HIVST service program. Our previous implementation evaluation study recruited alters directly in the platform but did not explore strategies to improve such recruitment [15]. This study provided an additional monetary incentive for index MSM to leverage their social networks and peer influence to distribute HIVST and encourage HIVST results returning from alters. It indicated that monetary incentives significantly improved HIVST secondary distribution among Chinese MSM. This strategy reached more MSM who had never received an HIV test before. Financial incentives have been applied across the HIV care continuum and have been found effective in improving uptake of HIV testing [16,31]. Incentives have also been used in other contexts to decrease risk behaviors and confront underlying social and structural vulnerabilities, such as poverty [32]. A previous study found that financial incentives increased HIVST uptake among male partners of pregnant women in Malawi [33]. Our findings further indicate that the monetary incentive approach is a powerful strategy to improve HIV social network–based HIVST among MSM.

In this study, a deposit was required for index individuals to acquire testing kits. The results may differ if implemented the program without this requirement. Specifically, the effect of monetary incentives could be either larger or smaller depending on whether monetary incentives substitute or complement any motivational effects of the deposit. Deposits could be an independent source of motivation if they acted as a commitment, whereby, after making a deposit, indexes would be motivated by so-called "sunk cost" effects [34]. The marginal effect of incentives could be more prominent if requiring a deposit is already a strong source of motivation (thus leaving less room for financial incentives to have notable effects). The effects of a monetary incentive could be smaller without the deposit, however, if the motivational effects of monetary incentives and deposit reinforce one another. Besides, deposits may potentially hinder the participation willingness among those who may be loss averse (89 MSM (21%, 89/417)) consented but an unpaid deposit with unknown reasons), it did ensure the overall performance of HIVST and secondary distribution. For instance, it may facilitate the likelihood to return results from testers, which can further benefit linkage to care and treatments where needed. Deposits may also prompt index participants to order an appropriate number of HIVST kits to reduce waste of HIVST kits or overtesting.

Our study also indicated that monetary incentives alongside a peer referral approach could also increase the secondary distribution of HIVST, motivating more alters for testing. This was consistent with the literature on peer referral of HIV self-test kits [35]. Our study tapped into the complexity of social networks in the internet era and leveraged digital technology by offering an opportunity for index MSM to send the ordering links to alters and, in turn, increasing testing coverage by encouraging more people within their social network for testing. However, our study results also suggested that online peer referral worked in conjunction with monetary incentives. Online peer referral alone had a suboptimal effect of motivating alters. We

speculated that monetary incentives were the key contributors to secondary distribution. Nevertheless, due to the incentivized secondary distribution of HIVST, which might potentially cause privacy concerns and limit the flexibility of alters, the strategy of online peer referral further decentralized power to alters of index MSM and allowed them more freedom to determine when, where, and who to test with. Simultaneously, index MSM also had more options, both in-person or online peer referral, for distributing HIVST services when their social contacts were reluctant to accept an HIVST test kit due to confidentiality or other concerns caused by in-person contact.

Additionally, our economic evaluation demonstrated that, compared to standard secondary distribution, small monetary incentives to index MSM lead to more alter testers and new alter testers getting tested, therefore lowering the average costs for each alter tested and new alter. Meanwhile, the cost for one additional tester was the smallest in SD-M arm when compared with the standard secondary distribution arm. Our findings suggested that from the health provider's perspective, SD-M could be an attractive use of health resources to expand HIVST among MSM. However, adding monetary incentives was more expensive than nonincentivized secondary distribution to identify an alter tester with a new HIV diagnosis. Given the low numbers in the trial, there is uncertainty surrounding this finding. Our economic data may help inform HIVST secondary distribution interventions in similar contexts. For those focusing on expanding HIVST using secondary distribution methods in resource restricting settings, adding a small monetary incentive to motivate index individuals holds promise to further amplify the value of distributing HIVST within their social networks. However, the generalizability of the economic findings in this study should be further examined by sensitivity analysis.

This integrated digital network–based self-testing approach may also hold promise in self-care interventions among key populations, such as other STI self-testing or self-collection. A qualitative study conducted in southern China among MSM demonstrated that their HIV/syphilis testing behaviors and preferences are associated with multilevel factors related to available testing technologies, stigma, service providers, and testing environments [36]. Our approach was built into an existing real-life gay community–led, social network–based HIVST service program, ensuring the provision of MSM-tailored and friendly health services. Index MSM in this process are not only healthcare seekers but also health providers who may potentially introduce HIV-related knowledge or offer HIVST kits to members within their social networks. The decentralized nature of the secondary distribution of HIVST enables index MSM to get tested and distribute kits outside clinical settings. Therefore, they contribute to cross the boundaries between institutions and the community, which is an innovative and effective approach to extend the leverage of social networks and reach marginalized and stigmatized populations [37]. COVID-19 exacerbated the profound inequalities running through the society [2]. Our integrated approach can be a promotion of decentralized sexual health services, addressing the inadequate response to HIV/STI services among marginalized populations in LMICs. Moreover, our subgroup analysis also suggested interventions differently impacted MSM with disparate characteristics, including age, sexual orientation, sexual orientation disclosure, HIV testing history, and condomless sex within 3 months. It provided critical implications for future implementation and optimization of the interventions.

This study had several limitations. First, our study implementation covered the lockdown period (January to May 2020) in China, during which HIV services were disrupted. How this has impacted our study results was uncertain. However, given the randomization of study participants and similar exposure to COVID-19 public health measures, we anticipated the impacts on study outcomes between the 3 groups would be similar. Further, during this period, HIVST was the only HIV testing option during the lockdown period, and facility-based testing services were unavailable. Our approach helped maintain the HIV care

continuum by providing HIVST via a digital platform and expanded HIV testing coverage among MSM during a period when many HIV testing sites closed. Second, this study provided no evidence that the 2 interventions group were any better than the control group in finding alters newly diagnosed with HIV. This may be related to our small sample size. Additionally, some people who were self-tested as HIV positive may refuse to return their testing results. However, the returning rate in our study was high (between 93% and 97% across different study groups), and its impact may be small. Third, although our study had a high follow-up rate within 3 months, the loss of follow-up data may be biasing the results. However, the loss of follow-up rate was comparable between the 3 groups, and participants who were not followed up had similar social demographic characteristics as those who completed the 3-month follow-up survey. Fourth, unnecessary testing may occur during the experiment because multiple kits might be ordered, and index participants were expected to use and distribute them within 3 months. However, the original intent that permitted the individuals to order multiple kits was to encourage index participants to apply for more kits and distribute more kits to people in their social network. Finally, this digitally implemented program may neglect those who have limited internet accessibility or digital literacy. The index participants might reach this group through social network kits distribution, but future studies are needed to address this limitation.

In conclusion, this study enhanced our understanding of the impacts of small monetary incentives and an additional online peer referral option in improving the reach of secondary distribution of HIVST. These findings have important implications for expanding HIVST among key populations using social network–based strategies in the digital era. This study provides foundational evidence for effective novel strategies to bridge the gap in HIV testing as an essential element of the HIV prevention and care cascade. Furthermore, this form of social network–based digital approach may apply to other public health research, especially in the era of COVID-19.

## Supporting information

**S1 Table. Characteristics of the index participants regarding 3-month follow-up survey completion, 2019 to 2020 (*N* = 309).**
(DOCX)

**S2 Table. Summary of the total number of ordered kits and actual kits distribution by index participants in each arm, 2019 to 2020 (*N* = 309).**
(DOCX)

**S3 Table. Characteristics of the alters in China, 2019 to 2020 (*N* = 269).**
(DOCX)

**S4 Table. Cost of the 3 arms of the trial by fixed and variable costs (USD, 2020).**
(DOCX)

**S1 Fig. Cost of the 3 arms of the trial by fixed and variable costs (USD, 2020).**
(TIF)

**S2 Fig. Cost breakdown of the 3 arms of the trial (USD, 2020).**
(TIF)

**S1 File. CONSORT 2010 Checklists. CONSORT, Consolidated Standards of Reporting Trial.**
(DOCX)

**S2 File. CHEERS Checklists.** CHEERS, Consolidated Health Economic Evaluation Reporting Standards.
(DOCX)

**S3 File. Study protocol.**
(DOCX)

**S4 File. Study protocol amendments.**
(DOCX)

**S1 Data. Deidentified study data.**
(ZIP)

**S2 Data. Cost data.**
(XLSX)

## Author Contributions

**Conceptualization:** Yi Zhou, Ying Lu, Yuxin Ni, Dan Wu, Jason J. Ong, Joseph D. Tucker, Sean Y. Sylvia, Fengshi Jing, Xiaofeng Li, Shanzi Huang, Guangquan Shen, Chen Xu, Fei Zou, Weiming Tang.

**Data curation:** Yi Zhou, Ying Lu, Yuxin Ni, Dan Wu, Xi He, Jason J. Ong, Joseph D. Tucker, Fengshi Jing, Xiaofeng Li, Shanzi Huang, Chen Xu, Yuan Xiong, Yongjie Sha.

**Formal analysis:** Yi Zhou, Ying Lu, Yuxin Ni, Dan Wu, Jason J. Ong, Joseph D. Tucker, Sean Y. Sylvia, Fengshi Jing, Guangquan Shen, Mengyuan Cheng, Fei Zou, Weiming Tang.

**Funding acquisition:** Weiming Tang.

**Investigation:** Yi Zhou, Ying Lu, Yuxin Ni, Xi He, Xiaofeng Li, Shanzi Huang, Chen Xu, Yuan Xiong, Yongjie Sha, Hongbo Jiang, Wencan Dai, Liqun Huang, Wenhua Mei, Weiming Tang.

**Methodology:** Yi Zhou, Ying Lu, Yuxin Ni, Dan Wu, Xi He, Jason J. Ong, Joseph D. Tucker, Sean Y. Sylvia, Shanzi Huang, Guangquan Shen, Chen Xu, Yongjie Sha, Mengyuan Cheng, Hongbo Jiang, Wencan Dai, Fei Zou, Cheng Wang, Bin Yang, Weiming Tang.

**Project administration:** Xi He, Xiaofeng Li, Yongjie Sha, Wenhua Mei, Weiming Tang.

**Resources:** Yi Zhou, Xi He, Shanzi Huang, Junjie Xu, Hongbo Jiang, Wencan Dai, Liqun Huang, Cheng Wang, Bin Yang, Wenhua Mei, Weiming Tang.

**Software:** Fengshi Jing.

**Supervision:** Yi Zhou, Ying Lu, Yuxin Ni, Dan Wu, Xi He, Joseph D. Tucker, Sean Y. Sylvia, Yongjie Sha, Junjie Xu, Hongbo Jiang, Wencan Dai, Liqun Huang, Cheng Wang, Bin Yang, Wenhua Mei, Weiming Tang.

**Validation:** Ying Lu, Yuxin Ni, Dan Wu, Xi He, Jason J. Ong, Joseph D. Tucker, Sean Y. Sylvia, Fengshi Jing, Xiaofeng Li, Shanzi Huang, Guangquan Shen, Yuan Xiong, Mengyuan Cheng, Junjie Xu, Hongbo Jiang, Fei Zou, Cheng Wang, Wenhua Mei, Weiming Tang.

**Visualization:** Ying Lu, Yuxin Ni, Chen Xu, Yuan Xiong, Mengyuan Cheng.

**Writing – original draft:** Yi Zhou, Ying Lu, Yuxin Ni, Dan Wu.

**Writing – review & editing:** Yi Zhou, Ying Lu, Yuxin Ni, Dan Wu, Xi He, Jason J. Ong, Joseph D. Tucker, Sean Y. Sylvia, Fengshi Jing, Xiaofeng Li, Shanzi Huang, Guangquan Shen,

Chen Xu, Yuan Xiong, Yongjie Sha, Mengyuan Cheng, Junjie Xu, Hongbo Jiang, Wencan Dai, Liqun Huang, Fei Zou, Cheng Wang, Bin Yang, Wenhua Mei, Weiming Tang.

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
