## [Editor Report · Decision Letter 0]

5 Jul 2021

Dear Dr Tang, 

Thank you for submitting your manuscript entitled "Monetary Incentives and Peer Referral in Promoting Digital Network-Based Secondary Distribution of HIV Self-Testing among Men Who Have Sex with Men in China: A Three-arm Randomized Controlled Trial" for consideration by PLOS Medicine.

Your manuscript has now been evaluated by the PLOS Medicine editorial staff and I am writing to let you know that we would like to send your submission out for external peer review.

Please re-submit your manuscript within two working days, i.e. by Jul 07 2021 11:59PM.

Kind regards,

Callam Davidson

Associate Editor

PLOS Medicine

---

## [Decision Letter · Decision Letter 1]

1 Sep 2021

Dear Dr. Tang,

Thank you very much for submitting your manuscript "Monetary Incentives and Peer Referral in Promoting Digital Network-Based Secondary Distribution of HIV Self-Testing among Men Who Have Sex with Men in China: A Three-arm Randomized Controlled Trial" (PMEDICINE-D-21-02857R1) for consideration at PLOS Medicine. 

[LINK]

In light of these reviews, we will not be able to accept the manuscript for publication in the journal in its current form, but we would like to consider a revised version that addresses the reviewers' and editors' comments. We cannot make any decision about publication until we have seen the revised manuscript and your response, and we plan to seek re-review by one or more of the reviewers. 

We hope to receive your revised manuscript by Sep 22 2021 11:59PM. Please email us (plosmedicine@plos.org) if you have any questions or concerns.

We look forward to receiving your revised manuscript. 

Sincerely,

Callam Davidson, 

Associate Editor 

PLOS Medicine

plosmedicine.org

In the interest of brevity, please consider shortening your title to ‘Monetary Incentives and Peer Referral in Promoting Secondary Distribution of HIV Self-Testing among Men Who Have Sex with Men in China: A Randomized Controlled Trial’.

The Data Availability Statement (DAS) requires revision - note that a study author cannot be the contact person for the data. Please direct data requests to a non-author institutional point of contact, such as a data access or ethics committee.

Line 37: Please update to ‘Conclusions’

Please trim the ‘Methods and Findings’ section of your abstract – the section should be a single paragraph containing the following: 

* The study design, population and setting, number of participants, exact dates during which the study took place, length of follow up, definition of intervention and control states, who was blinded to the group allocation, and main outcome measures.

* Please also provide the number in each group, state that analysis was intention to treat, quantify the main results (with 95% CIs and p values), and include the important dependent variables that are adjusted for in the analyses. Please also include a summary of adverse events if these were assessed in the study.

Citations should be in square brackets, and preceding punctuation, throughout the manuscript.

Please update the first line of the legend for Table 1 to ‘Age data are presented as mean (SD). All other data are presented as n (%)'.

The final figure provided (page 45 – diagram demonstrating principles of standard secondary distribution) does not appear to have a title and is not referenced in the main text. 

There are some discrepancies between your original trial registration (https://www.chictr.org.cn/com/25/historyversionpuben.aspx?regno=ChiCTR1900025433) and the manuscript, in terms of inclusion criteria and blinding). Can you please provide the reasoning behind these discrepancies? 

In the discussion, please consider discussing the concept of patients acting as providers - or extending the boundaries of the "health system" to include communities - an innovative and important approach to extending the reach and leverage of social networks, especially to stigmatized populations.

Please remove the ‘Financial Support’ section from your main text but include this information in your ‘Financial Disclosure’ statement in the submission form. In the event of publication, your form responses will be published as metadata. Please also remove the ‘Declaration of Interests’, ‘Data sharing’, and ‘Author Contributions’ sections from the manuscript main text and ensure that all the information is captured in your responses to the submission form. 

Please correct ‘Reference’ to ‘References’ on page 21. 

Please include the analysis plan, with any amendments, in the Supporting Information to be published with the manuscript if accepted.

Thank you for completing a CONSORT checklist. Please update the checklist to use section and paragraph numbers, rather than page numbers (as these may change during the revision process).

Please report your economic analysis according to the CHEERS statement and provide the relevant completed checklist - https://www.equator-network.org/reporting-guidelines/cheers/

Comments from the reviewers:

Reviewer #1: In this clearly presented, well-designed randomized trial, the authors test two additional strategies to amplify the impact of HIVST and peer testing for MSM in China: financial incentives and online peer referral. This presents relevant outcomes, including testing uptake overall and for first time testers and HIV and syphilis seroreactivity as well as costs. There are minor gaps that should be addressed prior to publication to strengthen this otherwise excellent trial.

1. Preferred terminology is "assigned male at birth" instead of "born biologically male". Do you know what proportion of the study population identified as cis-men versus trans-women or gender non-conforming?

2. The authors use a deposit in all arms of their trial, which is a separate mechanism of a financial incentive (contract) and has its own mechanistic influence on behavior. Why was this chosen? This should be added to the introduction and discussion section, along with a discussion of how this might impact trial results. 

3. The authors note that the number of alters per index might be artificially lowered because of the maximum of 5, 5, and 10 alter slots in each arm. Add details on what percentage of indexes in each arm maxed out the total number of alters allocated to further describe and quantify this potentially artificial ceiling. 

4. I am not familiar with the use of bootstrapping for statistical tests where data are not normally distributed (rather than using a non-parametric test or utilizing regression techniques that have more relaxed assumptions about normality). This should be seen by a statistical reviewer if not addressed in the initial review. 

5. Costing analysis: This is very important and I commend the authors for including it alongside the primary trial results. Table 4 has all of the relevant details for an incremental cost per X analysis, and yet all that is presented in the text is the cost per person tested or cost per person diagnosed, which is not the preferred metric for comparing models. The incremental cost per additional person tested is preferable and present in Table 4 but not discussed. 

6. Recruitment drop off: Any information on the drop off from 415 to 309? What do we know about the 106 who did not consent? What data are there to support that the deposit might be a barrier? This is a substantial fraction of the otherwise eligible population and would need to be described further. 

7. Line 247 how did the authors get 65 results from 58 unique alters? And same for line 250. This is unclear

8. Line 304 "average cost per tester was $24.54." Is this for index clients or all clients collapsed together? Which costs were the largest drivers? Which were the estimates most sensitive to? 

9. Discussion: all of the trial arms include a peer network approach, but what distinguishes third arm is the additional VIRTUAL/ONLINE peer referral instead of in person. This is not well described or clarified in this paper. This should be revised throughout to clarify that peer referral was present in all arms but the third arm had a virtual referral as well. 

10. Lines 343-347: how did this speculation compare to observation from the data?

11. Table 3 is fascinating and should be interpreted further in the results and discussion. Effects more pronounced in older, non-disclosed, and higher risk men. 

12. Figure 2 error bars needed

Reviewer #2: The paper describes a trial of two HIV self-test secondary distribution strategies among men having sex with men in China: incentives and online peer referral. It is an important study given the need to improve uptake of testing among key populations. There are however methodological weaknesses that call for caution in the interpretation of the findings:

Major comments

1. Deposit refunds were only given upon upload of test results. This could have forced index participants to over test so they could get reimbursed - there were 150 tests from 102 control participants, showing unnecessary testing and potential wastefulness of the intervention. Related to this, in lines 384-389, it does not seem convincing that the restriction on number of kits ordered was actually a limitation as it seems people ordered more than they could distribute

2. In the peer referral arm an index could distribute to/refer more people (ten) compared to five in the other arms. This makes results on mean numbers of testers incomparable with the other arms 

3. There were baseline differences by arm in participant characteristics. Please comment on potential for confounding

Minor comments

Abstract

4. Consider the rewording the definition of alters because in the abstract it sounds like an alter is the whole network/group, rather than one individual

5. Although it is clear in the manuscript, in the abstract it is unclear where self-test results were uploaded (line 16)

6. From the abstract it is not immediately clear what is being referred to as the digital approach

Manuscript

7. Page 13 lines 81-82: it is not clear what "may crowd them out" means

8. Line 158 - not clear whether the index received an additional 20rmb to the one described in line 157

9. Line 182 - Syphilis comes from nowhere without having been described in the methods nor any of the earlier sections. How and where were syphilis tests done?

Reviewer #3: Included in the uploaded file 

Reviewer #4: I confine my remarks to statistical aspects of this paper. Unfortunately, I think the statistical analysis was not correctly chosen; fortunately, the data are there to fix this pretty easily.

**Main issue**

The main issue is that the dependent variables are counts. Therefore, count regression models should be used. The authors could try Poisson regression but will likely want to move to negative binomial regression. as the assumptions of Poisson are rarely met. Simply taking confidence intervals and comparing them isn't adequate. A regression model would also allow the use of control variables such as age. The economic analysis should probably use OLS regression or maybe quantile regression.7u

p. 3 lines 28-30 (and also in the main text): Instead of giving mean and sd, please give median and IQR. Clearly, these variables are highly skewed and negative values are impossible. In addition, tables of the number of alters by group should be given (something like:

 Control SDM SDMPR

0

1

2

3

etc.

**More minor issues**

Starting on p 9 line 177: Remove each use of "mean". The outcome variable is the number of alters, not the mean number of alters. A variable should exist for each observation; here, each index person has a number of alters

Peter Flom

[LINK]

---

## [Decision Letter · Decision Letter 2]

5 Nov 2021

Dear Dr. Tang,

Thank you very much for submitting your manuscript "Monetary Incentives and Peer Referral in Promoting Secondary Distribution of HIV Self-Testing among Men Who Have Sex with Men in China: A Randomized Controlled Trial" (PMEDICINE-D-21-02857R2) for consideration at PLOS Medicine. 

Your paper was evaluated by an associate editor and discussed among all the editors here. It was also discussed with an academic editor with relevant expertise, and sent back to independent reviewers. The reviews are appended at the bottom of this email and any accompanying reviewer attachments can be seen via the link below:

[LINK]

In light of the remaining issues identified by the reviewers, we will not be able to accept the manuscript for publication in the journal in its current form, but we would like to consider a revised version that addresses the reviewers' and editors' comments. Obviously we cannot make any decision about publication until we have seen the revised manuscript and your response, and we plan to seek re-review by one or more of the reviewers. 

We hope to receive your revised manuscript by Nov 26 2021 11:59PM. Please email us (plosmedicine@plos.org) if you have any questions or concerns.

We look forward to receiving your revised manuscript. 

Sincerely,

Callam Davidson, 

PLOS Medicine

plosmedicine.org

Please provide details of who funded the study in your Financial Disclosure (the information removed from the previous version ought to be captured as part of your response to the submission form rather than removed entirely). 

Please define the abbreviations MSM and HIVST in your Author Summary.

Please ensure your Author Summary can be understood without the need to consult the main text (e.g. briefly define your intervention groups and avoid jargon where possible). 

In the Author Summary, under 'What Do These Findings Mean' bullet point 2, please correct 'effective' to 'effectively'

Please cite both your CONSORT and CHEERS checklists in your methods so readers know they can be found in the supplementary materials (currently reads 'available on request').

In the flow diagram (Figure 1), please indicate the number of individuals in each group analyzed in the ITT analysis.

Please update your CONSORT and CHEERS checklists to use section name and paragraph number (e.g. Discussion, Paragraph 2) rather than page number - page numbers change frequently during the revision process.

All information in your 'Declarations' file (supplementary material) should either be captured as part of your responses to the submission form (Declarations of Interests, Data availability, Author contributions) or should be relocated to the end of the main text (Acknowledgments). 

Comments from the reviewers:

Reviewer #1: Plos Med

In this revision, the authors have made substantial changes to the manuscript to address reviewer comments. There are a number of typos, lack of clear and precise language, and remaining inaccuracies that detract from this revision. In addition, a few additional comments would be important to address prior to publication.

1) The wording is unclear in several sections that have been rewritten, including a) the description of the deposit mechanism in the discussion, b) the description of the virtual/online peer referral in the third arm with peer referral. 

2) The authors responded to the question about whether the number of alters per index might be artificially lowered because of the maximum of 5, 5, and 10 alter slots in each arm. They added Supplementary table 1 and a narrative interpretation. However, Supplementary table 1 only shows the 5+ category in the bottom half of the table about the number of test kits distributed, rather than ordered, which prohibits assessing how many ordered >5 kits. Then, the number of kits distributed was higher in the SD-M-PR arm (as per the table), so the interpretation is inaccurate: "the upper limit of the number of kits order[ed] might not be a limitation of alters' motivation". This section should be revised for data completeness and accuracy.

3) The abstract was fully revised to remove much of the primary results of the paper in favor of a lot of details about the costing analysis; why were the primary results cut? 

4) The main point about the second arm including peer referral (in person) and the third arm including VIRTUAL or DIGITAL peer referral is still really missing in the paper and erroneously makes it seem as though "peer referral" (instead of "virtual peer referral") is the unique concept being tested in the third arm. This point was made nicely on page 8 in response to my prior comment, but then throughout the rest of the manuscript the same clarity is missing. 

5) Page 16 of the marked up manuscript, line 430 says "similarly" to compare two sets of outcomes in which one has an association and the other does not; is there a typo in the interpretation? 

6) Page 16 of the marked up manuscript, most of the absolute value changes and differences between arms have been removed, leaving only the relative changes (IRR). The absolute differences are important, even if only means without bootstrapped 95%Cis. 

7) The titles for Table 2 and 3 are mixed in the text. Subgroup analyses are Table 3, not 2. 

8) The directionality of the association of disclosure of sexual orientation and tested alters is different depending on the comparison groups; the text should reflect this mixed directionality and does not. 

9) I would ask the authors to re-interpret their cost-effectiveness section of the text, especially the wording related to more or less cost-effective. The table is correct but doesn't show the SD-M-PR arm compared to the control arm to support the text interpretation. 

Reviewer #2: Authors have put effort in responding to comments, with improved clarity. There are still aspects related to study design and execution, as well as interpretation, that remain unclear. Specific comments are below:

Major comments

1. The response to Comment 1 from Reviewer 2 still does not address the difference in numbers of tests among index participants and the number of index participants, and the likely over-testing. It would be important to spell out over what time/period the assessments of testing uptake were made for each index. Surveys were conducted three months after randomisation, were assessments of testing uptake also done over three months, in which case unnecessary testing was possible - there were more index testers than index participants, e.g. there were 150 tests received from 103 SD-M index participants?

2. Related to above, in response to comment 7 from reviewer 1, authors state that there were discrepancies related to people who uploaded results more than once. It would be expected that the study methods would be able to correct such duplication.

3. The response to Reviewer 3 on allocation concealment does not seem to address the question. How was allocation concealment done? Who told the participants what arm they were in (and how was this done) and how was the computer-generated code concealed from study staff? Was this all done electronically without need for staff interaction? The methods of recruitment and allocation are unclear

4. Minor comment: Page 11 Author summary: Under the paragraph "What do these findings mean", the term 'digital based programs' comes out of nowhere as the preceding paragraphs haven't mentioned that the programs are digital based

Reviewer #4: The authors have addressed my concerns and I now recommend publication.

Peter Flom

[LINK]

---

## [Decision Letter · Decision Letter 3]

21 Dec 2021

Dear Dr. Tang,

Thank you very much for resubmitting your manuscript "Monetary Incentives and Peer Referral in Promoting Secondary Distribution of HIV Self-Testing among Men Who Have Sex with Men in China: A Randomized Controlled Trial" (PMEDICINE-D-21-02857R3) for consideration at PLOS Medicine. 

Your paper was evaluated by an associate editor and discussed among all the editors here. It was also discussed with an academic editor with relevant expertise, and sent back to independent reviewers. The reviews are appended at the bottom of this email and any accompanying reviewer attachments can be seen via the link below:

[LINK]

In light of these reviews, I am afraid that we will not be able to accept the manuscript for publication in the journal in its current form, but we would like to consider a revised version that addresses the reviewers' and editors' comments. We cannot make any decision about publication until we have seen the revised manuscript and your response, and we plan to seek re-review by one or more of the reviewers. 

We hope to receive your revised manuscript by Jan 11 2022 11:59PM. Please email us (plosmedicine@plos.org) if you have any questions or concerns.

We look forward to receiving your revised manuscript. 

Sincerely,

Callam Davidson, 

PLOS Medicine

plosmedicine.org

Please clearly state and define the primary outcomes of the study in the abstract.

In the abstract, please include effect sizes and 95% CI for the primary outcomes (those found in the lowermost rows of Table 2).

Please remove the sentence 'The study design and reporting of the results were followed the CONSORT 2010 guidelines' from the abstract.

Please include the numbers of new HIV infections in the abstract.

Please remove p-values from Table 1.

Please amend 'newly testers' in the title of Figure 2.

Throughout the manuscript, please report relevant p-values after 95% CI. 

At line 191, please make sure it is clear to readers that all participants were required to pay the deposit, not just some of them (or alternatively, please let me know if I have that wrong). 

Comments from the reviewers:

Reviewer #1: The authors did a strong and detailed revision to all of my prior comments. Brief requested changes below, after which this article is suitable for publication:

1. Table 2 column headers use the phrase "by each index participant" but it seems to be the total number from ALL index participants. Check and revise throughout as appropriate.

2. The footnote of Table 2 says "Zero inflated negative binomial regression was used to calculate IRR for outcomes "number of unique alters tested" and "number of newly-tested alters for HIV". Logistic regression was used to calculate OR for outcome "Number of alters identified with HIV-reactive results". Why were different regression methods used for these analyses?

3. The new CEA interpretation has a sentence, "For programs with a budget of more than $7688·30 and a cost-effectiveness threshold of no less than $17.73 per tester..." Where diid these numbers come from? The $17.73 does not match numbers given in the table and the total budget seems to be driven by the sample size in the study, not a population. The cost-effectiveness thresholds are somewhat arbitrary; are there any data about other interventions that are more or less costly per person tested that have been adopted to ground these estimates in?

Reviewer #2: The manuscript now reads better and is clearer; reviewer comments have been addressed satisfactorily.

[LINK]

---

## [Decision Letter · Decision Letter 4]

10 Jan 2022

Dear Dr. Tang,

Thank you very much for re-submitting your manuscript "Monetary Incentives and Peer Referral in Promoting Secondary Distribution of HIV Self-Testing among Men Who Have Sex with Men in China: A Randomized Controlled Trial" (PMEDICINE-D-21-02857R4) for review by PLOS Medicine.

I have discussed the paper with my colleagues and the academic editor and it was also seen again by one reviewer. I am pleased to say that provided the remaining editorial and production issues are dealt with we are planning to accept the paper for publication in the journal.

[LINK]

We look forward to receiving the revised manuscript by Jan 17 2022 11:59PM.   

Sincerely,

Callam Davidson, 

Associate Editor 

PLOS Medicine

plosmedicine.org

Requests from Editors:

Line 22: Please update the phrasing here to more clearly define your primary outcomes in the abstract. For example, “The primary outcomes were the mean number of motivated alters who have photo-verified self-testing per index in each arm, the proportion of first-time HIV testing among alters in each arm, and the proportion of alters with a positive HIV testing result in each arm. These were assessed using zero-inflated negative binomial regression.” Please adjust phrasing as required to ensure consistency between the abstract and the main text. 

Line 25: Please indicate that these results are mean ± standard deviation.

Line 35: Please expand this sentence to state the total number of HIV positive participants, and the proportion of first-time HIV testing among alters in each arm (per the outcomes listed in the protocol). Consider including a sentence relating to the third outcome, e.g., “there were similar proportions of alters newly testing positive in the three groups”.

Please mention the refundable deposit in the abstract. 

Line 75: Please include standard deviation in the Author Summary.

Line 84: In the interest of brevity, please remove this bullet point (I feel the following bullet conveys a similar point). 

Line 246: Please update ‘Our primary outcomes included to ‘Primary outcomes were …’.

Line 256: Please check the numbering of your supplementary file contents – the protocol amendments are located in S4 not S3 (please check throughout).

Line 269: Please note CONSORT discourages baseline comparisons. Please ensure these have been consistently removed throughout the manuscript (beyond removing the p-values from Table 1), from e.g. lines 315-16.

Line 278: ‘HIV-reactive results’.

Line 302: Please report <1% rather than 0%.

Line 322: Please add a space between ‘returned’ and ‘209’.

Line 344: As above, this should be Supplementary Table 2 not 1 (please check throughout).

Line 363: Please remove the comma after the full stop.

Lines 376-377: Please rephrase this sentence for clarity (the wording at line 353-355 is more clear).

Line 475: ‘would be a cost-effective strategy’.

478: Please delete the word ‘however; from this sentence.

Lines 521-523: Please provide a supplementary table with the data to support this statement (participants who were not followed up had similar social demographic characteristics as those who had completed the three-month follow-up survey) and add a reference to it in your Results section. Please re-number all supplementary tables as required. 

Table 2: Unless I am missing it, I cannot see any data to which the footnote labelled ** applies (p<0.001).

Supplementary Table 3: Please include the full excel sheet of cost items in your supplementary material as a study author cannot be the point of contact for data requests.

Comments from Reviewers:

Reviewer #1: The authors have made many revisions to this manuscript in this most recent submission, some of which add additional confusion, unfortunately. 

1) Table 2: while the authors stated in their response letter that the headers should refer to "by all participants" rather than "by each participant", but the header was revised to read "Total number of unique alters motivated by each index participant". Recommend revising to have each  all or "total number of unique alters motivated by participants". As is, it is unclear. (Also recommend a spell check to catch typos like "umber" in lieu of "number" in header). 

2) Table 4 has been flipped in orientation for some reason and the resultant table is not a traditional CEA format and is less clear. If the authors prefer to retain as is for a particular reason, that's fine, but it is an unnecessary change that does not aid in clarity.

3) The authors state in their response letter that they've removed the thresholds for cost-effectiveness, but this partially remains in the manuscript, along with cost-effective language. Recommend a careful re-read of every section of the manuscript that deals with the health economics evaluation in full with attention to prior review comments to make changes consistent and accurate throughout instead of partially changed and with some retained errors in reach revision. I did not re-read every section in careful detail to catch each of these errors beyond the first one noted.

[LINK]

---

## [Editor Report · Decision Letter 5]

19 Jan 2022

Dear Dr. Tang,

Thank you very much for re-submitting your manuscript "Monetary Incentives and Peer Referral in Promoting Secondary Distribution of HIV Self-Testing among Men Who Have Sex with Men in China: A Randomized Controlled Trial" (PMEDICINE-D-21-02857R5) for review by PLOS Medicine.

The remaining issues that need to be addressed are listed at the end of this email. Please take these into account before resubmitting your manuscript:

We hope to receive your revised manuscript within 1 week. Please email me (cdavidson@plos.org) if you have any questions or concerns.

We look forward to receiving the revised manuscript by Jan 26 2022 11:59PM.   

Sincerely,

Callam Davidson, 

Associate Editor 

PLOS Medicine

plosmedicine.org

Requests from Editors:

Thank you for providing the cost data excel sheet. Please confirm this spreadsheet does not contain any data that could compromise participant privacy.

Please update the last sentence of the S4 Table legend from 'available on request' to 'available in the supplementary materials'.

Please delete “(statistician)” at line 9 (abstract).

At line 9 (abstract), please make that “index participants”.

At line 23, we suggest some rewording for clarity: “Primary outcomes were the differences in mean number of unique motivated alters with photo-verified self-testing by index participants in each arm; and the mean number of newly-tested alters motivated by index participants in each arm.”

At line 41, please make that “tested HIV positive”.

At line 50, please make that “Limitations include …”

At line 51, please move the trial registration number to a separate line at the end of the abstract. Please move the phrase regarding ITT analysis earlier in the abstract, e.g., to line 27.

At line 427 (first paragraph of Discussion), please add a sentence, say, to summarize the trial’s findings. 

Please use “PLoS” consistently in the reference list.

---

## [Editor Report · Decision Letter 6]

21 Jan 2022

Dear Dr Tang, 

On behalf of my colleagues and the Academic Editor, Dr Elvin Geng, I am pleased to inform you that we have agreed to publish your manuscript "Monetary Incentives and Peer Referral in Promoting Secondary Distribution of HIV Self-Testing among Men Who Have Sex with Men in China: A Randomized Controlled Trial" (PMEDICINE-D-21-02857R6) in PLOS Medicine.

When making the formatting changes, please also make the following updates:

* Please update your methods (lines 255-256) to refer to 'mean' rather than 'total' numbers (for consistency with the way you define outcomes in your abstract).

* Delete 'Total number of' from all column headers in Table 2 (as the columns actually contain numbers, means, and measures of risk).

PRESS

Sincerely, 

Callam Davidson 

Associate Editor 

PLOS Medicine